# Technological Windows of Opportunity for Russian Arctic Regions: Modeling and Exploitation Prospects

**Vera P. Samarina [1], Tatiana P. Skufina [1], Diana Yu. Savon [2] and Svetlana S. Kudryavtseva [3,\*]**

1   Kola Science Center of the Russian Academy of Sciences, 184209 Murmansk Region, Russia;
    samarina_vp@mail.ru (V.P.S.); skufina@iep.kolasc.net.ru (T.P.S.)
2   Department of Industrial Management, National University of Science and Technology (MISiS),
    119991 Moscow, Russia; savon.dy@misis.ru
3   Department of Logistics and Management, Kazan National Research Technological University,
    420015 Kazan, Russia
*   Correspondence: sveta516@yandex.ru; Tel.: +7-902-7152968

**Abstract:** The problems of exploitation of technological windows of opportunity are of particular scientific and practical interest in terms of the development of Russia's national economy, and the Arctic region, which has a strong mineral and raw materials potential, is important in terms of its use for achieving the technological and national security of the Russian state. Considering this, the study of the theoretical and methodical aspects of the development of emerging technological windows of opportunity is important and relevant for the regions of the Russian Arctic zone. The purpose of this study is to assess the potential and reserves for exploitation of the emerging technological windows of opportunity during the deployment of a new technological order by mobilizing material and human capital in the Arctic regions. Methodological tools for the study of this problem included dynamic series analysis, structural analysis, comparison, description, descriptive statistics, cross-correlation analysis, production function model and its visualization. An analytical review of scientific publications, a set of tools and methods of research, allowed to obtain the following scientific results: A significant variability in the contribution of science-intensive and high-tech industry to the formation of gross value added in the Arctic region has been revealed; meanwhile, we can note stable dynamics of the contribution of the Arctic economy to the gross domestic product (GRP) of Russia as a whole. There is a steady excess of the productivity index over the Russian average, which can be regarded as a potential for growth of high-tech components of labor in the development of the economy of the Arctic region. There is a negative statistically significant relationship between the share of the gross regional product of the Arctic in the Russian GRP and the share of gross value added (GVA) of science-intensive products in the Arctic GRP, which can be regarded as a factor preventing the exploitation of the emerging technological windows of opportunity. The construction of a model of production function of technological windows of opportunities for the Arctic zone of Russia pointed to the presence of potential in the exploitation of emerging technological windows in the Arctic zone of Russia in the development of human capital through the activation and use of high labor productivity, creating high-performance jobs. The results of the study, its findings and its proposals can be used in the development, monitoring and implementation of state federal and regional programs and projects aimed at improving the level of technology and science intensity of production in the Arctic zone, improving its competitiveness, which is highly important for the national economy.

**Keywords:** technology windows of opportunities; science-intensive products; high-tech products; productivity; development investments; cross-correlation function; production function

## 1. Introduction

The problem of development of the Arctic zone has been becoming particularly important and relevant over the past few years. These aspects are associated, firstly, with

development and implementation of the Sustainable Development Goals (Sustainable Development Goals 2021), secondly, with the need to reduce differentiation in the level of socio-economic development of Russian regions, and thirdly, in the search for new opportunities for innovative, scientific and technological development of border areas in order to achieve national and technological security. Since 2015, Russia has been implementing the state program "Social and Economic Development of the Arctic zone of the Russian Federation" (Government Programs Portal 2021). One of the key goals of this program is the development of science and technology and the increase in the efficiency of using the resource base of the Russian Arctic zone and the continental shelf of the Russian Federation in the Arctic. We believe that these prospects for scientific and technical, technological and science-intensive production, with the support of the state and business, can be considered as emerging technological windows of opportunity when estimating technological orders and management paradigms in the modern innovative economy. The key aspect of using the emerging technological windows of opportunity is the formation of an advanced scientific and technical basis and technologies in promising directions for the Arctic zone—oil and gas and industrial engineering—which will facilitate the production of competitive high-tech products and will provide a significant reduction in the technological lag of the Russian economy behind the world level.

The research problem is that at present, there is no comprehensive analysis of assessing the impact of indicators of economic and technological development on changes in the macroeconomic parameters of the Arctic regions. In this regard, it is relevant to study the development trends of the Arctic region within the framework of a unified systematic assessment, taking into account the relationship of the main macroeconomic parameters. The study involves the use of modeling tools, in particular, correlation-regression analysis, trend analysis and a production function model. The hypothesis of the research is that for the Arctic regions, despite the high capital intensity and energy intensity of economic development, the priority in the opening technological windows of opportunities belongs to human capital, as the driver of the program.

The Arctic zone is one of the hard-to-reach regions, which makes it difficult to study and introduce innovations in order to achieve sustainable development. On the other hand, the Arctic region has a significant natural resource potential, the study, disclosure and use of which can become a key factor in ensuring technological and environmental safety not only for the Russian Federation, but also for the global ecosystem as a whole. In this regard, the problem of studying the prospects for the use of opening technological windows of opportunities when changing technological structures for the development of the Arctic regions is considered as significant and relevant.

At the end of 2020, the volume of investments in fixed capital for the development of the Arctic territory amounted to 1528.4 million rubles, which is 10% of the total volume of Russia-wide investment. Of these, the share of own funds was 53.5% and the share of obtained funds was 46.5%. In the types of investments in fixed assets, the largest share of investment was in buildings, 55.3%, machines and equipment, 23.8%, and the objects of intellectual property, 4.7% (Rosstat 2021). At the same time, the total volume of the state program for the development of the Arctic zone during the period of its implementation will be about 7 billion rubles from budgetary sources until 2025.

Considering this, we believe that the need to explore the emerging technological windows of opportunities for the development of the Arctic regions of Russia in the modern innovation economy is becoming particularly important and relevant.

The problem of development of the Arctic, including the economic, social and environmental aspects of the topic, is reflected in such areas of scientific research as the Arctic air flows (You et al. 2021), Arctic water management (Alkire et al. 2021), management of the Arctic carbon footprint (Amon 2021), mountain seismicity of the Arctic regions (Shebalin et al. 2020), Arctic ecosystem (Csapó et al. 2021) and Arctic climate change (Tseng 2021; Durner et al. 2009). However, these issues are mainly focused on the study

of the environmental problems of the Arctic zone, while the economic and technological developments of the Arctic zone regions within them are addressed indirectly.

The study of the possibilities of emerging technological windows is noteworthy, in particular: "green windows of opportunity" (Dai et al. 2020), "green technologies" (Bas and Oliu 2018; Zhou et al. 2020), etc., however, similar to the study of the development of the Arctic, environmental focus prevails in scientific research on this problem. In this study, we will understand, via technological windows of opportunity, the prospects for using the latest scientific achievements for the development of high-tech activities that open up for production when changing technological structures.

The issues of achieving sustainable development are also important for the Arctic region, as reflected in research areas such as the regional and international problem of achieving the Sustainable Development Goals (Tambovceva et al. 2019; Fahed and Daou 2021), spatial development (Popović et al. 2021), modernization of management concepts (Skufina et al. 2019; Spinosa and Doshi 2021; Novoselova et al. 2020), development of Arctic frontier territories (Samarina et al. 2018), interrelation between macroeconomic indicators and Sustainable Development Goals (Cook and Davidsdottir 2021) and others. The environmental component also dominates these studies.

The problems of economic development of the territories in the face of changing technological systems are represented by such areas of research as the management of intellectual capital in the innovation economy (Kudryavtseva and Shinkevich 2015), cluster management technologies for changing technological orders (Dyrdonova 2016), socio-economic differentiation of regions in the era of the fourth industrial revolution (Vertakova et al. 2017), factors of innovation in prospective sectors of the economy (Klimenko et al. 2018), business systems in the context of transformational economy (Klochko and Brizhak 2019), digital ecosystem modernization (Shkarupeta et al. 2020), etc. Despite the wide range of considered problems during the change of technological orders, the authors do not view them in the context of individual territories, in particular, the Russian Arctic zone.

The theory on the issue of searching and expanding opportunities for Russian arctic regions is presented in the works of the following authors: a multi-criteria approach to land-use planning in northern Quebec (Grandmont et al. 2012), thaw settlement in soils of the Arctic Coastal Plain (Pullman et al. 2007), comparative estimates of Kamchatka territory development in the context of northern territories of foreign countries (Shelomentsev et al. 2014), comparative analysis of regional development of Northern Territories ( Shelomentsev et al. 2015), development of the Arctic regions of the Russian Federation (Voronina 2020), development problems of the Arctic Circle (Koch et al. 2020), climate change (Czerniawska and Chlachula 2020), effects of experimental warming in the Arctic (Davenport et al. 2020), etc.

Thus, despite the extensive coverage of Arctic development issues, as the "windows of opportunity" in achieving the goals of sustainable development of the Arctic zone, we believe it is necessary to supplement the existing areas of research by studying technological and human resources using the emerging technological windows of opportunities in the change of technological orders for the Arctic regions. This provision predetermined the choice of the topic of the study, the tools used and the structure of the article.

## 2. Materials and Methods

The research methodology consists in generalizing scientific and theoretical approaches to the study of opening technological windows of opportunities when changing technological structures in the development of the Arctic zone. The research methodology is based on the following methodological principles: consistency, complexity, integrity, scientific validity of conclusions and statistical reliability of the results. The object is the ecosystem of the Arctic region, and the methods are economic and mathematical modeling methods. The methodology is based on the use of the following approaches in the management of the Arctic region: institutional, project and system.

Conceptually, the research methodology included the collection of aggregated relative indicators of the technical and economic development of the Arctic region, published on the Rosstat website, their systematization, analysis using trend models, the identification of patterns and the development of economic and mathematical models, allowing to describe the relationship between these indicators.

Methodological tools for the study include dynamic series analysis, structural analysis, comparison, description, descriptive statistics, cross-correlation analysis, production function model and its visualization. The statistical basis of the study was the official data of the Federal State Statistics Service, reflecting statistical information on the socio-economic development of the Arctic zone (Rosstat 2021).

In our study, the focus is on assessing the impact of the development of the level of technology and science-intensity of production in the Arctic zone on the increase in the gross value added generated in the region, as an indicator of the potential and prospects for increasing the competitiveness of the region, not only in the national but also in the global economic system.

In the first phase of the modeling using the toolkit of descriptive statistics, the statistical indicators for the parameters analyzed were evaluated, namely:

Y, the share of gross regional product (GRP) of the Arctic in the GRP of the Russian Federation, %.

X1, the share of gross value added (GVA) of science-intensive products in the Arctic GRP, %.

X2, the share of science-intensive products in shipped products of Arctic enterprises, %.

X3, the index of labor productivity in the Arctic, %.

The dynamic series of the indicators analyzed included data for the period of 7 years, from 2014 to 2020, as monitoring of indicators of socio-economic development in the Russian Arctic zone started in 2014.

Using the toolkit of descriptive statistics, the average and median values of the indicators, variance and scope of this sample of indicators were analyzed, which allowed to reveal the overall picture of the indicators of the technological and science intensity of the production in the Arctic zone, as well as its compliance to the law of normal distribution of values.

In the second phase of the modeling, the cross-correlation functions between the simulated indicators were calculated, taking into account the time lags, which allowed to determine the relationship between the resulting factor Y, the share of GRP of the Arctic in the GRP of the Russian Federation, and the explaining variables, X1–X3, when they were changing over a period of time.

In the third phase of the modeling, a production function model was proposed and produced, describing the potential for exploiting the emerging technological windows of opportunity for the Arctic zone through increased use of material capital, science-intensive production and human capital through productivity growth.

In general, the production function model is presented in the following form:

$$Y = A \times K^{\alpha} \times L^{\beta}$$

where:

A is a production function parameter,
K is capital costs,
L is labor costs,
Y is the share of Arctic GRP in the GRP of the Russian Federation, %,
$\alpha$ and $\beta$ are parameters of the production function.

Using the existing statistical base on the indicators of the development of the Arctic region, we transformed the production function model according to the following parameters in relation to the topic of this study:

Y, the share of gross regional product (GRP) of the Arctic in the GRP of the Russian Federation, % (result).

K, the share of science-intensive products in shipped products of Arctic enterprises, % (capital expenditure).

L, the labor productivity index in the Arctic, % (labor costs).

The proposed econometric model is a modification of the production function model, which makes it possible to assess the effect of changes in the gross regional product under the influence of changes in the costs of labor and capital indicators, which makes it possible to use it for analytical purposes when studying the opening technological windows of opportunity in the Arctic region. Based on the coefficients of elasticity of the production function model, this effect can be estimated, taking into account the value of the parameters for the independent variables included in the production function model. The indicators participating in the model are brought to a comparable form by means of logarithms.

## 3. Results

### 3.1. Descriptive Analysis of the Technological Level of the Russian Arctic Zone

One of the key indicators for assessing the regional economic development level is the gross regional product. For the period 2014–2020, the average value of the share of the Arctic GRP in the GRP of the Russian economy was 5.63%, and in 2020, it was 6%. It is possible to assess the technological level of the economy of the Arctic zone using the indicators of the generated gross value added in the science-intensive sector of the economy in the region's GRP. Therefore, at the end of 2020, the value of this indicator was 6.6%, and on average for 2014–2020, it was 7.01% (the average for the Russian Federation is 19.2%), hence, there is a significant (2.7 times) difference in the level of technological production of the Arctic zone and the average Russian value. Both indicators are characterized by a linear trend, however, for the share of high-tech and science-intensive products in the Arctic GRP, the trend line has a negative slope, which is associated with a decrease in the value of this indicator in 2016–2019 (Figure 1).

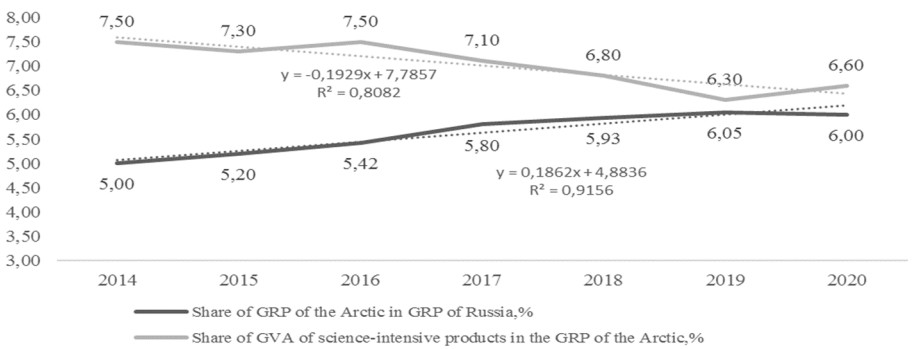

**Figure 1.** The dynamics of gross value added (GVA) of the science-intensive industry and the gross regional product (GRP) of the Arctic zone (compiled by the authors).

Less than 1% is the share of high-tech and science-intensive products in the total volume of shipped products of enterprises located in the Arctic zone (0.05% in 2020). Meanwhile, we should note that the value of this indicator on average in the Russian economy was also insignificant, 1.2%.

Among the positive factors influencing the exploitation of technological windows of opportunity in the Arctic is the labor productivity indicator, which on average was higher than in the Russian Federation in 2014–2020, 103.2% versus 101.5%, respectively (Figure 2).

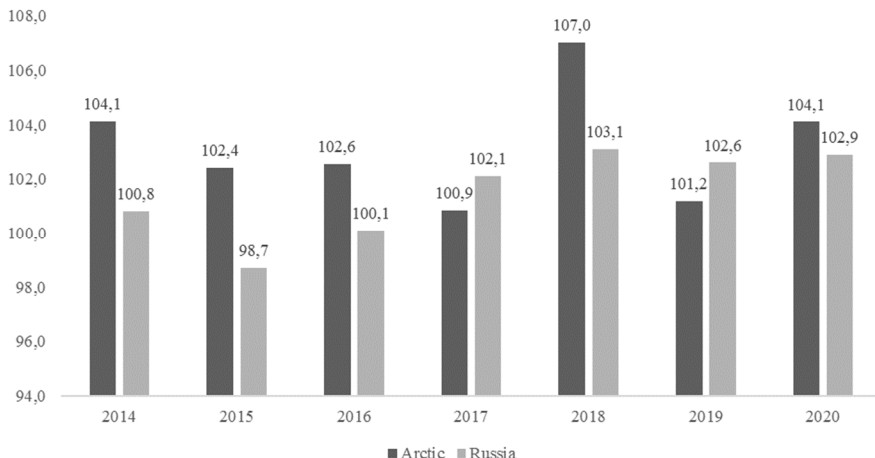

**Figure 2.** Labor productivity index dynamics, percentage (compiled by authors).

In general, the distribution of the technology and science-intensiveness of the production of the Arctic zone was in line with the law of normal distribution of values, which enables us to use them for further analysis. However, we should note that a slight unevenness in the trends of indicators in 2014–2020 was present, for example, for GRP and GVA indicators, there is a slight left-sided asymmetry (average value is less than the median), and for the indicators of the share of science-intensive products in shipment and productivity indexes, there is slight right-sided asymmetry (average value is greater than the median) (Table 1).

**Table 1.** Descriptive statistics.

| Ind. | Mean | Median | Minimum | Maximum | Range | SD |
|------|------|--------|---------|---------|-------|-----|
| Y | 5.63 | 5.80 | 5.00 | 6.05 | 1.05 | 0.42 |
| X1 | 7.01 | 7.10 | 6.30 | 7.50 | 1.20 | 0.46 |
| X2 | 0.07 | 0.05 | 0.05 | 0.17 | 0.12 | 0.04 |
| X3 | 103.18 | 102.55 | 100.85 | 107.00 | 6.15 | 2.11 |

Thus, the regions of the Arctic zone are characterized by a small value of GVA formed by science-intensive products in GRP, however, there is a high growth potential for the use of human capital in the science-intensive and high-tech industry.

*3.2. Cross-Correlation Functions of the Development of the Technological Level in the Russian Arctic Zone*

The calculation of cross-correctional functions between the indicators of technological development of the Arctic zone allowed to establish the following. There is a negative statistically significant relationship ($p \leq 0.05$) between the gross regional product of the Arctic in the GRP of Russia (Y) and the share of GVA of science-intensive products in the GRP of the Arctic (X1): the cross-correlation coefficient was negative, –0.885, which can be explained by the presence of a negative trend for the first indicator. There was no time lag between these indicators (Figure 3).

However, cross-correction analysis did not allow to establish a statistically significant high correlation ($p \geq 0.05$) between the share of the gross regional product of the Arctic in the GRP of Russia (Y) with the share of science-intensive products in shipped products of Arctic enterprises (X2) and the index of labor productivity in the Arctic (X3) ($p \geq 0.05$). Thus, at this stage of development, the Arctic GRP is formed by non-technological types of production, which require increased use of the scientific and technological component in the expansion and development of the industry of the Arctic region (Figure 4).

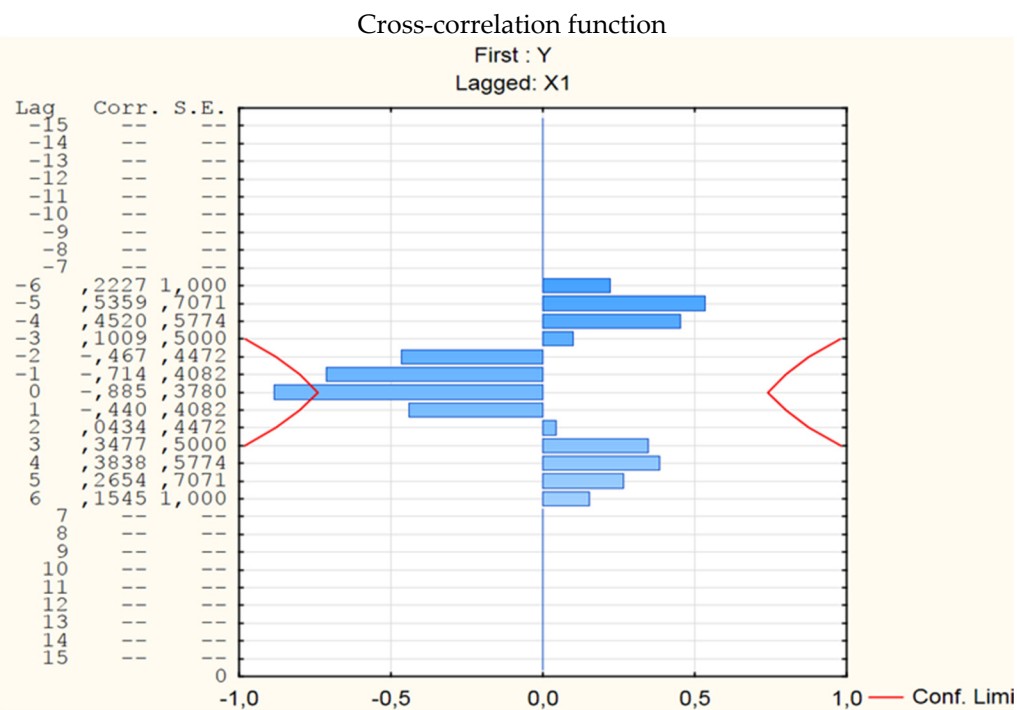

**Figure 3.** Cross-correlation function for GRP and GVA of the science-intensive production of the Russian Arctic zone (calculated by the authors).

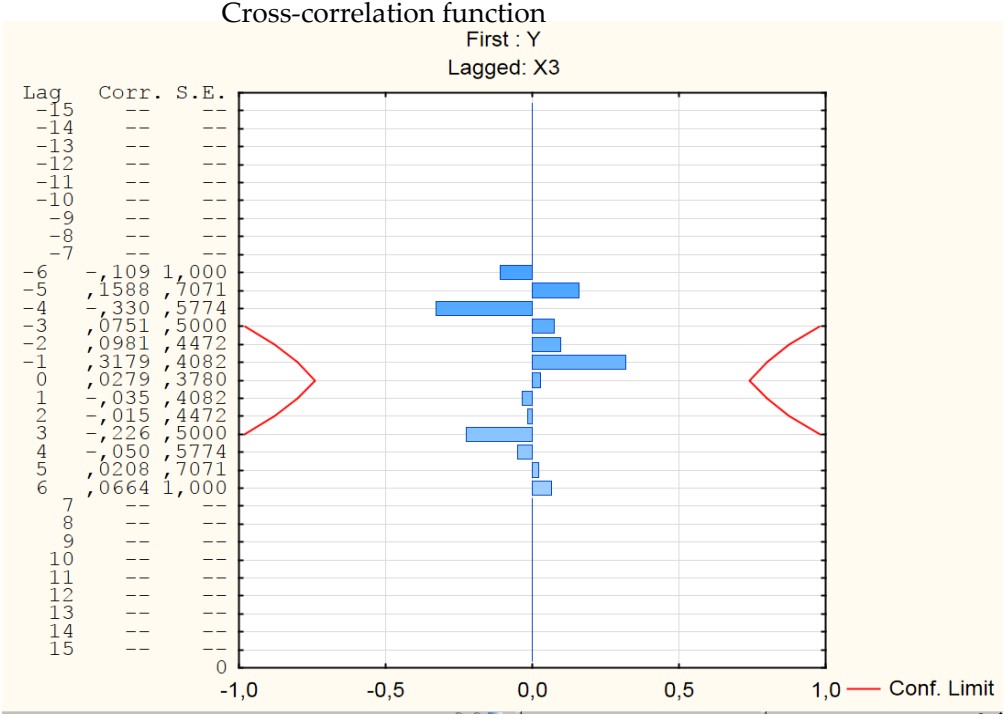

**Figure 4.** Cross-correlation function for GRP, share of science-intensive products in shipment and productivity index of the Russian Arctic zone (calculated by authors).

### 3.3. Modeling the Production Function of Technological Windows of Opportunity for the Russian Arctic Zone

Due to the high negative correlation between the share of gross regional product in the GRP of Russia (Y) and the share of GVA of science-intensive products in the GRP of the Arctic (X1), in order to avoid the effect of multi-collinearity in the model of production

function of technological windows of opportunities for the Russian Arctic zone, we will include the following parameters for modeling:

Y, the share of gross regional product (GRP) of the Arctic in the GRP of the Russian Federation, % (result).

X2, the share of science-intensive products in shipped products of Arctic enterprises, % (capital expenditure).

X3, the labor productivity index in the Arctic, % (labor costs).

The following model of the production function was obtained as a result of the simulation:

$$Y = 1.38 \times X1^{(-0.11)} \times X2^{(0.23)}.$$

The coefficient of determination ($R^2$) of the model was 53%, and the standard error of the model was 6%, which is the average quality of the model. However, the complexity of modeling is due to the fact that there are no longer-term dynamics in terms of indicators for the Arctic zone of the Russian Federation.

As we can see, capital expenditures are characterized by a negative impact on the exploitation of the emerging technological windows of opportunity in the regions of the Arctic zone considering its negative elasticity ratio, which was −0.11, while the model demonstrates the positive effect of labor costs in the exploitation of the emerging technological windows of opportunity, where the elasticity ratio was 0.23 and by its module it is 2 times the value of capital costs. Accordingly, the 1 percent increase in the labor productivity index provides an increase in the share of gross regional product of the Arctic in the Russian GRP by 0.23 percentage points.

Visualization of the production function model of the technological windows of opportunity for the Russian Arctic zone is presented in Figure 5.

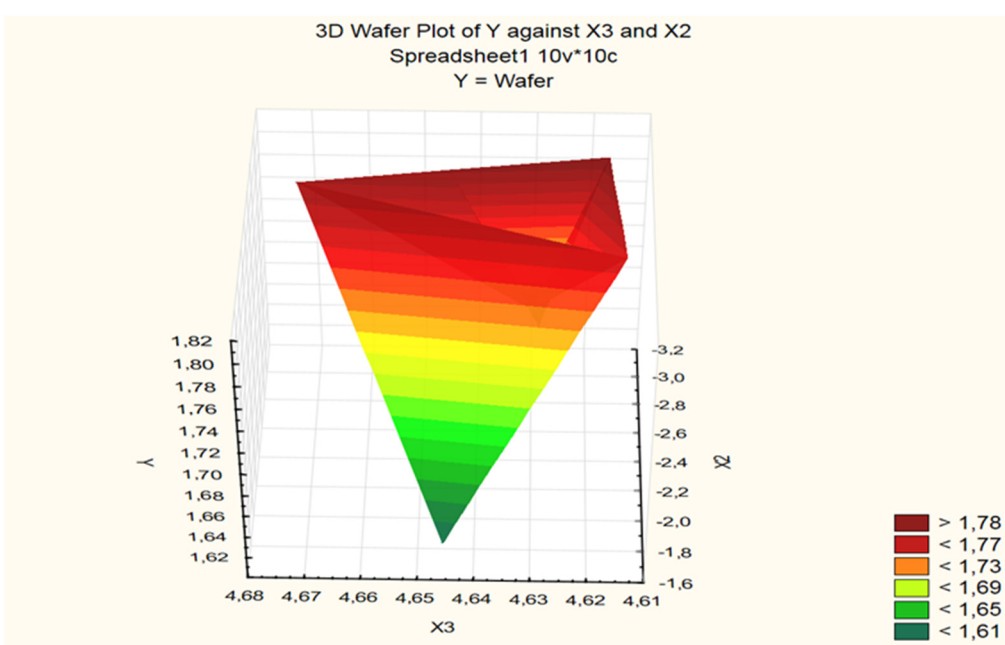

**Figure 5.** A 3D diagram of the surface of the production function model (calculated by the authors).

Thus, based on the results of the modeling, we can conclude that the potential for exploiting the emerging technological windows of opportunity in the Russian Arctic zone belongs, first of all, to the development of human capital through activation and use of high labor productivity and creation of high-performance jobs, which will allow to start the development of new science-intensive and high-tech industries, ensuring an increase in the regional gross value added of the science-intensive sector of the economy and GRP.

At the same time, the Arctic zone, as a developing region, is carrying out modernization by involving, first of all, human capital. However, the obtained conclusion suggests that the capital-intensive component, expressed in the growth of science-intensive products, has a negative trend, which negatively affects the mobilization and building up of human potential in the region. Consequently, this may indirectly indicate the low return of human potential capable of creating science-intensive products, which increases the gross value added in the region.

We believe that in order to solve this problem, it is necessary to create and develop an innovative infrastructure in the region, and design an appropriate institutional structure aimed not only at increasing human potential, but also transforming it into capital through the growth of knowledge-intensive types of economic activity.

## 4. Conclusions

The study of the prospects for the development of the emerging technological windows of opportunities in the Russian Arctic zone allowed us to draw the following conclusions:

1.  There is considerable variability in the contribution of science-intensive and high-tech industries to the formation of gross value added in the Arctic region. At the same time, there is a stable dynamic of the contribution of the Arctic economy to the formation of the GRP of Russia as a whole.
2.  There is a steady growth of the productivity index over the average Russian indicators, which can be considered as the potential for growth of high-tech components of labor in the development of the economy of the Arctic zone.
3.  A negative statistically significant relationship has been identified between the share of the gross regional product of the Arctic in the Russian GRP and the share of GVA-intensive products in the Arctic GRP, which may be regarded as a factor preventing the use of opportunities of the emerging technological windows.
4.  Building a model of production function of technological windows of opportunities for the Russian Arctic zone pointed to the presence of potential in the exploitation of emerging technological windows of opportunity in the Russian Arctic zone in the development of human capital by activating and using high labor productivity, and creating high-performance jobs.

Based on the results of the study, it is possible to propose a set of measures exploiting the emerging technological windows of opportunity for the Arctic regions:

-   Inclusion of cross-indicators into the program of socio-economic development of the Arctic zone of the Russian Federation, allowing to assess and analyze the dynamics of the relation between indicators of material and human capital development.
-   Development of regional and industry sub-programs and projects of development of the region's human capital as a key driver of technological development.
-   Development of basic projects to expand the range of high-tech production facilities of the existing technological order for the Arctic regions, implemented based on the principles of public–private partnership.
-   Implementation of industrial and social infrastructure projects aimed at achieving balance of interests in the development of material and human capital in the region.

The theoretical significance of the study lies in the generalization and systematization of institutional theory, project management theory and the theory of systems for managing the development of the Arctic region based on the use of opening technological windows of opportunities when changing technological structures. In addition, the methodology and tools presented in the article can be used as an initial methodological base for further research in this area in order to improve the efficiency of management of the Arctic territory.

We believe that the results of the study, its conclusions and its proposals can be used in the development, monitoring and implementation of state federal and regional programs and projects aimed at improving the level of technology and science of produc-

tion in the Arctic zone, increasing its competitiveness, which is highly important for the national economy.

**Author Contributions:** Conceptualization, V.P.S., T.P.S., D.Y.S. and S.S.K.; methodology, V.P.S., T.P.S. and S.S.K.; formal analysis, D.Y.S.; investigation, S.S.K.; data curation, V.P.S., T.P.S., D.Y.S. and S.S.K.; writing—original draft preparation, D.Y.S. and S.S.K.; writing—review and editing, V.P.S. All authors have read and agreed to the published version of the manuscript.

**Funding:** This research was funded by the Russian Science Foundation, Project No. 19-18-00025 (Arctic technology window of opportunity assessment), state assignment of the Federal Research Center "Kola Science Center of the Russian Academy of Sciences" No. AAAA-A18-118051590118-0 (methodology of evaluation).

**Institutional Review Board Statement:** Not applicable.

**Informed Consent Statement:** Not applicable.

**Data Availability Statement:** The article used data from open information sources: Rosstat https://rosstat.gov.ru (accessed on 23 May 2021).

**Conflicts of Interest:** The authors declare no conflict of interest.

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
