# Peer review of "Technological Windows of Opportunity for Russian Arctic Regions: Modeling and Exploitation Prospects"

_jrfm, doi:10.3390/jrfm14090400_

Round 1
Reviewer 1 Report
Dear Authors,
I see that you corrected the text according to my instructions. However it would be nice to mark in the text all the changes (in colour for instances).
In the figure 1 - you use russian language. Please translate it.
Author Response
Thank you! Agree. The article was edited according to the comments.

Reviewer 2 Report
The paper submitted for review represents a potentially interesting research problem, unfortunately it doesn’t fulfill all the standards of high-quality scientific papers.
The Introduction section generally lacks a well-defined research problem. What is the main objective, what do the authors want to study, with what methods (short presentation), what will be the added value of this study in relation to already existing studies/research in this area? Introduction should lead the reader from a general subject area to a particular topic of inquiry. It establishes the scope, context, and significance of the research being conducted by summarizing current understanding and background information about the topic, stating the purpose of the work in the form of the research problem supported by a hypothesis or a set of questions, explaining briefly the methodological approach used to examine the research problem, highlighting the potential outcomes your study can reveal, and outlining the remaining structure and organization of the paper (https://libguides.usc.edu/writingguide/introduction).
Lines 81-87: It seems that Arctic climate change and global warming are the same problem. If not, please precise.
Line 116: Effects with a lowercase letter
Materials and Methods section: What is missing here, is a comprehensive, clear explanation of the research methodology. The methods section describes actions to be taken to investigate a research problem and the rationale for the application of specific procedures or techniques used to identify, select, process, and analyze information applied to understanding the problem, thereby, allowing the reader to critically evaluate a study’s overall validity and reliability. The methodology section of a research paper answers two main questions: How was the data collected or generated? And, how was it analyzed? (Kallet, Richard H. "How to Write the Methods Section of a Research Paper." Respiratory Care 49 (October 2004): 1229-1232.)
The Results and Discussion section: This section lacks a scientific discussion of the results. The purpose of the discussion is to interpret and describe the significance of your findings in light of what was already known about the research problem being investigated and to explain any new understanding or insights that emerged as a result of your study of the problem. The findings and their implications should be discussed in the broadest context possible and limitations of the work highlighted. Future research directions may also be mentioned. The discussion will always connect to the introduction by way of the research questions or hypotheses you posed and the literature you reviewed, but the discussion does not simply repeat or rearrange the first parts of your paper; the discussion clearly explain how your study advanced the reader's understanding of the research problem from where you left them at the end of your review of prior research. (Annesley, Thomas M. “The Discussion Section: Your Closing Argument.” Clinical Chemistry 56 (November 2010): 1671-1674.)
Lines 290-295: This is not a revealing conclusion. It applies to every less developed region.
Please revise your final Conclusions to make them understandable to a potential reader who is not familiar with econometric terms. Imagine that this article is given to a person responsible for implementing a development strategy for the Arctic region to read. What will such a person understand from it?
The paper requires some serious revisions before being published.
Author Response

(The authors gave the same response as above.)

Reviewer 3 Report
The paper is correct, and I understand that the originality lies in the study of potential for exploiting the emerging technological windows of opportunity in the Russian Arctic zone. However, the results could be defined as tautological, applicable to any other region to which the analysis could be applied.
It would be interesting to explain how the authors propose that, given the particularities of the region, it is possible "the development of human capital through activation and use of high labor productivity, creation of high-performance jobs, which will allow to start the development of new science-intensive and high-tech industries, ensuring an increase in the regional gross value added of the science-intensive sector of the economy and GRP" (lines 290-206).
That is, how the market by itself can attract and maintain human capital in a region such as the Arctic or the social and institutional conditions must be created so that these emerging technological windows of opportunity can be effective. The arguments in this sense are lacking for the work "can be used in the development, monitoring and implementation of state federal and regional programs and projects aimed at improving the level of technology and science of production in the 338 Arctic zone" (lines 336-338).Author Response
Thank you! Agree. The article was edited according to the comments.

Round 2
Reviewer 2 Report
It seems everything is fine, therefore I accept the current version of the paper for publication.
This manuscript is a resubmission of an earlier submission. The following is a list of the peer review reports and author responses from that submission.
Round 1
Reviewer 1 Report
Thank you for the opportunity to read the research materials.
The practical part of the article is well prepared. However, despite the fact that this document demonstrates the high practical importance of the problem, in its current form it cannot be accepted for publication.
Recommendations to authors.
- Finalize the theoretical description of the problem, as well as the analysis of previous studies in this area.
- Add research methodology. Now she's gone.
- In the conclusions, the authors should more reasonably prove the theoretical significance of the results obtained for the development of world science
- It is necessary to supplement the list of bibliographic sources with theoretical and empirical research, including works by Russian and foreign authors.
To supplement the theory on the issue of searching and expanding of opportunity for Russian arctic regions, it is recommended to study the following works:
Grandmont, K., Cardille, J.A., Fortier, D., Gibryen, T. (2012) Assessing land suitability for residential development in permafrost regions: A multi-criteria approach to land-use planning in northern Quebec, Canada. Journal of Environmental Assessment Policy and Management, 14 (1), art. no. 1250003. doi: 10.1142/S1464333212500032
Pullman, E.R., Jorgenson, M.T., Shur, Y. (2007)Thaw settlement in soils of the Arctic Coastal Plain, Alaska. Arctic, Antarctic, and Alpine Research, 39 (3), pp. 468-476. doi: 10.1657/1523-0430(05-045)
Shelomentsev A.G.,Kozlova O.A.,Terentyeva T.V., Bedrina Ye.B. 2014. Comparative estimates of Kamchatka territory development in the context of northern territories of foreign countries. Economy of Region, 2, рр.89–103. doi: 10.17059/2014-2-9
Shelomentsev, A.G., Kozlova, O.A., Mingaleva, Z.A., Bedrina, Y.B., Terentyeva, T.V. 2015. Comparative analysis of regional development of Northern Territories. Asian Social Science. 11(14), рр. 349-356 doi: 10.5539/ass.v11n14p349
Voronina E. 2020. Development of the Arctic regions of the Russian Federation: Drivers of greening. E3S Web of Conferences. 2441.9 art. no. 10051. DOI 10.1051/e3sconf/202124410051
Author Response
Dear expert, thank you for the appreciation of the article. All recommendations were taken into account and included in the article.
Reviewer 2 Report
Comments and Suggestions:
The topic of the article is important and relevant. But the applied methods and approaches require more careful study.Section 2. Materials and Methods The empirical model is not well founded.
The article makes a reference to the Cobb-Douglas two-factor production function.
The authors do not show how this can be converted to deduce the dependency
share of GRP of the Arctic in Russian GDP from selected factors.
Section 3. Results
The authors' conclusions are based on the results of a seven-year time series analysis (seven observations in total). This length of time series is not sufficient either for estimating cross correlations
or for reliably estimating the parameters of regression models. Therefore, the conclusions in subsections 3.2 and 3.3 do not seem to be valid.
Technical notes and typos
Line 62. the Arctic territory to 1 528.4 million ... (rubles, dollars?)
Line 144 Instead Y = A × Kα × Lβ, should be Y = A × K ^ α × L ^ β.
Line 228
The equation "Y = 1.38 × Х1-0.11 × Х2 0.23" is likely to contain typos. Also, there are no values for standard errors of coefficients, coefficient of determination, etc.
Author Response

(The authors gave the same response as above.)

Reviewer 3 Report
Dear Authors,
The paper provides an interesting point of view of the technological windows of opportunity for Russian arctic regions, however it needs minor improvement.
First of all, you should explain more the concept of “windows of opportunity” in the light of literature - if you use this term in the title.
In the abstract you use the abbreviations: GRP and GVA without the explanation.
Line 44 – Sustainable Development Goals should be written in capital letters.
I suggest to change Discussion part to Conclusion.
You can add the section: Discussion where you should refer to other studies in the literature about this topic and write more about the limitations of the study.
In the references, there are too large spacing between words (line 304, 306, 318, 331, 332).
Author Response

(The authors gave the same response as above.)

Round 2
Reviewer 1 Report
Thank you for taking into account the comments and remarks. However, the theoretical review should still be expanded and not limited only to those publications that were recommended. The article can be published.
Reviewer 2 Report
The research methodology needs to be significantly improved.It is not shown how the proposed econometric model can be derived from the production function.
The time series contains only seven observations.
The model indicated on line 263 does not comply with the further explanations (lines 268-275).